



# CH4Net: a deep learning model for monitoring methane super-emitters with Sentinel-2 imagery

Anna Vaughan[1], Gonzalo Mateo-García[2,3], Luis Gómez-Chova[3], Vít Růžička[4], Luis Guanter[5,6], and Itziar Irakulis-Loitxate[5,7]

[1] Computer Laboratory, University of Cambridge, UK

[2] Trillium Technologies Ltd., London, UK

[3] Image Processing Laboratory, University of Valencia, Valencia, Spain

[4] University of Oxford, Oxford, UK

[5] Universitat Politècnica de València, Valencia, Spain

[6] Environmental Defense Fund, Reguliersgracht 79, 1017 LN Amsterdam, the Netherlands

[7] International Methane Emission Observatory, United Nations Environment Program, Paris, France.

**Correspondence:** Anna Vaughan (av555@cam.ac.uk)

**Abstract.** We present a deep learning model, **CH4Net**, for automated monitoring of methane super-emitters from Sentinel-2 data. When trained on images of 21 methane super-emitters from 2017-2020 and evaluated on images from 2021 this model achieves a scene-level accuracy of 0.83 and pixel-level balanced accuracy of 0.77. For individual emitters, accuracy is greater than 0.8 for 17 out of the 21 sites. We further demonstrate that CH4Net can successfully be applied to monitor two superemitter locations with similar background characteristics not included in the training set, with accuracies of 0.92 and 0.96. In addition to the CH4Net model we compile and open source a hand annotated training dataset consisting of 925 methane plume masks.

## 1 Introduction

As a potent greenhouse gas responsible for approximately 25% of warming since the industrial revolution (Stocker, 2014; Varon et al., 2021) with rapidly increasing atmospheric concentrations (Tollefson, 2022), curbing methane emissions is an important step in combating the climate crisis. Anthropogenic emissions emanate from diverse sources, principally associated with livestock, agriculture, landfills, and the fossil fuel industry (oil and gas extraction and coal mining) (Saunois et al., 2020; Maasakkers et al., 2022). Of particular interest for rapid suppression of emissions are super-emitters, defined to be sources in the top 1% of global anthropogenic methane emitters, corresponding to an approximate flow rate of 25 kg/h (Zavala-Araiza et al., 2017). These sources contribute a substantial fraction of all methane emissions in the oil and gas sector (Alvarez et al., 2018), providing an opportunity to rapidly limit emissions with mitigation at a reasonable cost (Lauvaux et al., 2022).

Over the past five years, remote sensing instruments have been extensively utilised for detecting and monitoring super-emitters (Irakulis-Loitxate et al., 2022; Lauvaux et al., 2022; Varon et al., 2021; Maasakkers et al., 2022; Irakulis-Loitxate et al., 2021). To monitor these point sources, it is necessary to use point source imagers, instruments with a spatial resolution of less than 60 m (Jacob et al., 2022). In addition to this, the ideal instrument would also have global coverage, rapid revisit



time, and high spectral resolution in the 1700 and 2300 nm short wave infrared spectral windows where methane absorption is the strongest. Unfortunately, no currently available instrument has all of these desired characteristics.

Hyperspectral instruments, for example PRISMA and EnMAP, provide the most accurate concentration retrievals. However, these instruments suffer a significant limitation in that they require tasking and hence have limited data availability.

An alternative approach is to utilise multispectral imagery such as Sentinel-2 (Drusch et al., 2012) and Landsat-8 and 9
(Roy et al., 2014). These instruments have relatively rapid revisit time (approximately five days for Sentinel-2 and 16 days for Landsat at the equator) and high (20-30m) spatial resolution. They do, however, have significantly degraded spectral resolution compared to hyperspectral instruments, resulting in a lower sensitivity to methane and hence a stronger sensitivity to the background surface. Recent work has demonstrated successful detection and quantification of large plumes from Sentinel-2 imagery (Varon et al., 2021; Ehret et al., 2022; Irakulis-Loitxate et al., 2022). These approaches are based on temporal
differences and ratios between Sentinel-2 bands 11 (1560–1660 nm) and 12 (2090–2290 nm). Band 12 strongly overlaps with the methane absorption feature, while band 11 provides an estimate of the background at a relatively similar wavelength. Varon et al. (2021) present a series of approaches differencing between S2 bands 11 and 12 to quantify methane emissions. Their most successful approach quantifies emissions down to a rate of 3 t/h (tons of CH4 emitted per hour) by taking the difference of bands 11 and 12 comparing two consecutive passes, however remains sensitive to surface artefacts. Ehret et al. (2022) take a
similar approach projecting onto a time series of 30 previous images with two-stage linear regression and a manual verification step to identify the presence of false positives caused by surface artefacts. There are two significant limitations with these methods. The first and most important is that that they remain sensitive to surface artifacts, often requiring manual verification. The second is that a timeseries of images is required.

In this study, we take an alternative approach and train a machine learning model, CH4Net, to segment methane plumes from
a single image without the need for a time series of previous images, reference image or manual verification step. Machine learning has been successfully applied to segmenting plumes in hyperspectral data (Groshenry et al., 2022; Jongaramrungruang et al., 2022; Schuit et al., 2023), however, this methodology has not yet been applied to Sentinel-2 imagery as a sufficiently large dataset of verified plumes is unavailable. We first collect and annotate a dataset of methane plumes from known super-emitters in Turkmenistan (Irakulis-Loitxate et al., 2022), a semi-arid region with strong emissions providing the best-case scenario for
multispectral methane mapping. This is used to train a deep learning model to segment methane plumes from the background. We evaluate this model for a future time period for the training locations. In addition, we show that the model can successfully be applied to monitor a super-emitter at a new location in the same region unseen at training time. The aims of this paper are as follows:

1. Collect and label a machine learning dataset of methane plumes in Sentinel-2 imagery.

2. Develop an automated plume segmentation system. In contrast to existing works, this is a fully automated system that does not require a time series of Sentinel-2 images or identification of a reference image.

3. Apply this system to track emissions from a selection of known methane super-emitters during a future time period.

4. Quantify skill at new locations unseen at training time.



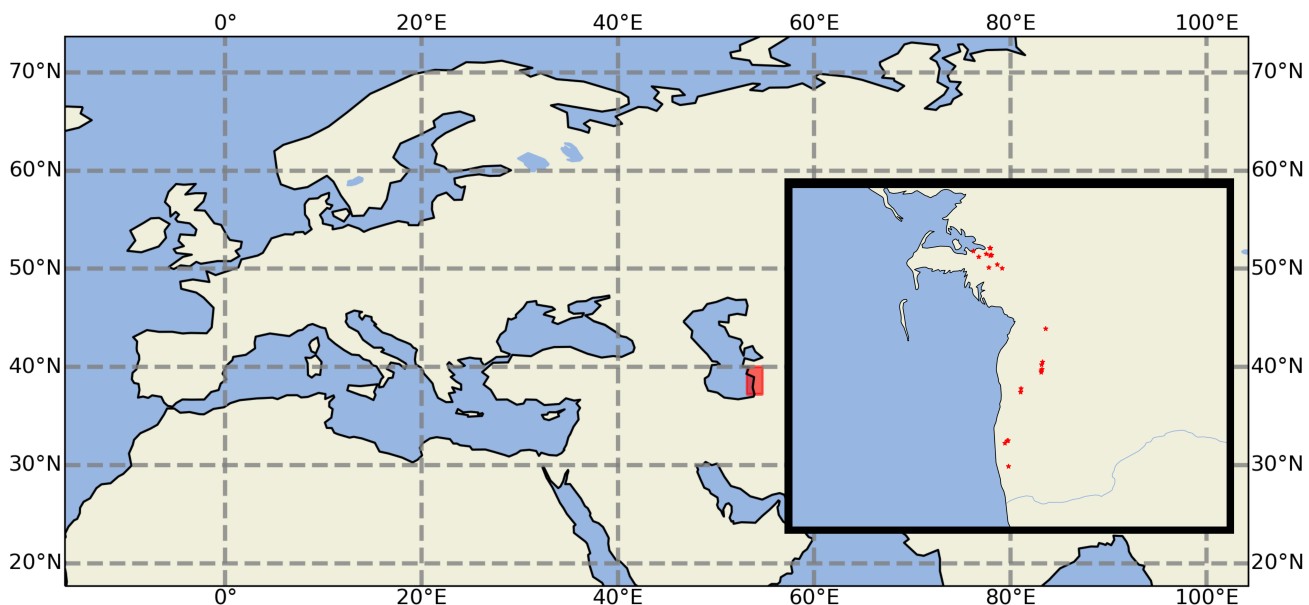

**Figure 1.** Locations of the 23 super emitters included in the dataset showing the study region shaded in red and precise locations (inset).

Section 2 presents an overview of dataset collection, the CH4Net architecture and training procedure. Results are presented in Section 3, with conclusions and a discussion in Section 4.

## 2 Methods

### 2.1 Dataset collection and processing

We first collect and manually annotate a dataset of methane plumes from Sentinel-2 images from 2017-2021 consisting of 10,046 0.01×0.01 degree images ( 200×200 pixels) from Sentinel-2 L1C scenes centred on 23 known super-emitter locations in Turkmenistan (Irakulis-Loitxate et al., 2022). Several locations identified are in close proximity to each other, and are combined into a single scene. For a map and complete list of included sites, see Figure 1 and Table 1. For each site all available images were downloaded using the Sentinel Hub API, each consisting of the 13 scaled and harmonized Sentinel-2 channels (Sinergise Ltd., 2023). Images containing clouds are deliberately not discarded to allow the model to learn a mapping robust to these features without the need for costly preprocessing steps. We note that the model output is therefore predicting whether a plume is visible in the scene or not; it is possible that an emission may be present but is covered by clouds. Cloudy scenes could easily be discarded if necessary for a particular application by applying a cloud detection model (Jeppesen et al., 2019; López-Puigdollers et al., 2021; Aybar et al., 2022).

We frame methane detection as a binary segmentation problem, where a pixel is classified as either 0, if not part of a plume, or 1, if part of a plume. To label the plumes, enhanced images were created for each time-step using the multi-band multi-



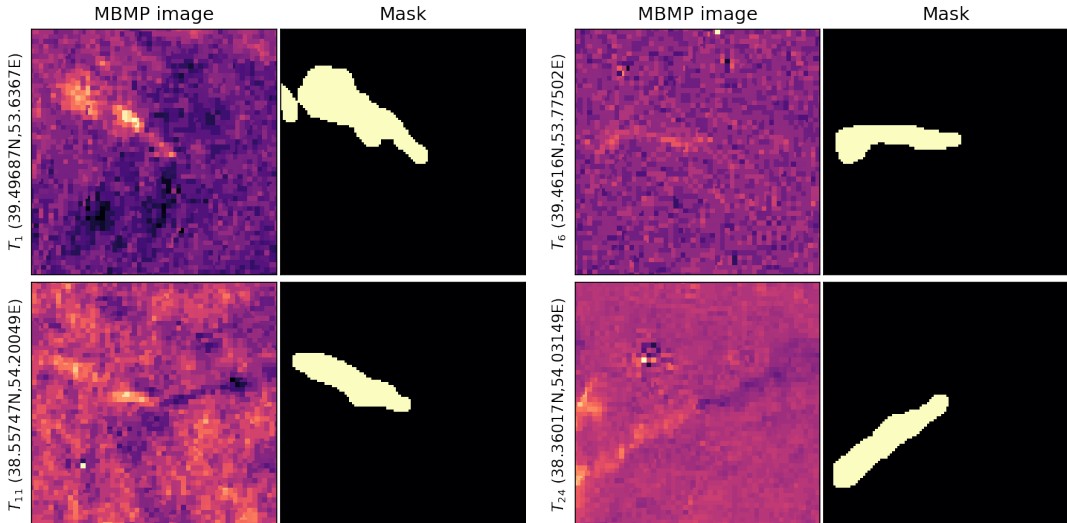

**Figure 2.** Examples of the MBMP images and corresponding hand annotated masks.

pass (MBMP) method developed by Varon et al. (2021). A clear-sky reference image was chosen for each location, with the multi-band multi-pass image given by

$$MBMP = \frac{cR_{12} - R_{11}}{R_{11}} - \frac{c'R'_{12} - R'_{11}}{R'_{11}}$$

where $R_{11}$ and $R_{12}$ are the raw Sentinel-2 band 11 and 12 observations for the current image, $R'_{11}$ and $R'_{12}$ are the raw Sentinel-2 band 11 and 12 observations for the reference image and $c$ ($c'$) is calculated by least-squares regression of $R_{11}$ against $R_{12}$ ($R'_{11}$ against $R'_{12}$) for all pixels. These images were used to manually identify and label the extent of the methane plumes for each time-step. For examples of the MBMP images and corresponding hand-labelled plumes, see Figure 2. It is emphasized that these MBMP images are used as an auxillary tool to guide annotation only and are not included as input predictors to the final model.

Each data point consists of the 13 Sentinel-2 bands interpolated to a common resolution of 10m together with the hand-labelled plume mask for a total of 925 scenes containing a plume and 9121 without. The resolution of 10m is chosen as adding the highest resolution RGB channels improves the model performance, so all data is interpolated to this resolution to avoid loss of information. This dataset is split into train (2017-2020; 7783 images with 659 containing plumes), and validation (2021; 2263 images with 266 containing plumes) sets. For the validation set, 266 negative images are randomly sampled to prevent the metrics being biased towards negative samples. In addition, we hold two locations out from the training set entirely to test



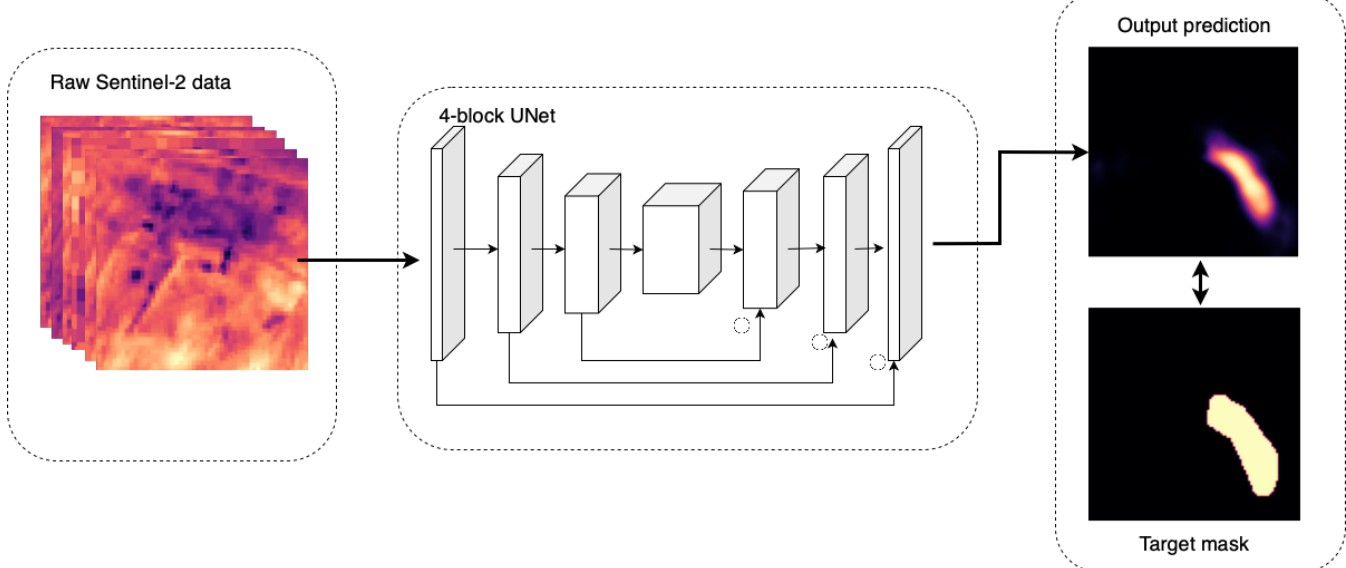

**Figure 3.** Schematic of the CH4Net model architecture showing the Sentinel-2 bands input to the UNet and probabilistic output compared to the hand-annotated mask.

the ability of the model to generalise to new sites. This is referred to as the held out dataset containing images over the two sites in 2021 (210 total images, 9 plumes).

## 2.2 Model architecture and training

The detection model uses a simple and flexible UNet architecture Ronneberger et al. (2015) consisting of 4 encoder blocks (2D convolution layer, batch norm, ReLU activation, 2D convolution layer, batch norm ReLU activation, maxpool) followed by four decoder blocks (transposed 2D convolution layer, 2D convolution layer, batch norm, ReLU activation, 2D convolution layer, batch norm ReLU activation) with skip connections between blocks of corresponding scale. Channel output dimensions for each of these blocks are $\{128, 256, 512, 512, 256, 128, 64, 128, 1\}$ with kernel sizes of 3 for all convolution layers and 2 for the max pooling layers. For a complete schematic of the model see Figure 3. This model takes the Sentinel-2 bands as input and outputs a pixelwise prediction of the probability of the pixel being part of a methane plume.

The UNet is trained on the training dataset described above with Binary Cross-Entropy loss, Adam optimisation (Kingma and Ba, 2014) and a learning rate of $1e-4$ for 250 epochs. As the dataset is unbalanced with significantly more negative than positive images, at each epoch $n$ negative images are randomly sampled, where $n$ is the total size of the positive image set. To prevent overfitting augmentation is applied by cropping a random 100x100 pixel scene from the larger image tiles. In order to investigate the optimal predictor set, the UNet is trained with both bands 11 and 12 only as predictors (11+12), and all bands (ALL).





## 3   Validation results

We first evaluate the skill of CH4Net at correctly identifying whether a given image contains a methane plume. This is referred to as scene-level prediction, as opposed to pixel-level prediction. For scene-level prediction the probabilistic predictions are transformed to a binary prediction by defining a methane plume as a contiguous region of greater than 115 pixels with probability greater than or equal to 0.25. The 115 pixel threshold is chosen as this is the size of the smallest plume contained in the training set, while the 0.25 threshold is selected to maximize the balanced accuracy score. A scene is classified as 1 (containing a plume) if such a feature is present and 0 otherwise.

Accuracy, precision, recall, false positive rate, and false negative rate for both the ALL and 11+12 experiments over the 2021 validation set are shown in the upper portion of Table 1. The model with all bands included as predictors outperforms that with only bands 11 and 12, indicating that other bands add value for methane detection, or for the reduction of false positives. Results over the validation set for the model with all bands included are accuracy 0.83, precision 0.82, recall 0.82, false positive rate 0.17 and false negative rate 0.18, indicating good skill at correctly determining between scenes with and without a plume.

**Table 1.** Scene and pixel level metrics over the validation dataset (year 2021) for CH4Net trained with the complete 13 band predictor set (ALL) and the bands 11 and 12 only predictor set (11+12).

| Scene level metrics | | |
|---|---|---|
| | ALL | 11+12 |
| Accuracy | 0.83 | 0.79 |
| False positive rate | 0.17 | 0.19 |
| False negative rate | 0.18 | 0.23 |
| Precision | 0.82 | 0.77 |
| Recall | 0.82 | 0.80 |
| Pixel level metrics | | |
| Balanced accuracy | 0.77 | 0.73 |
| Accuracy | 0.98 | 0.97 |

A more challenging task is to assess prediction skill at a pixel level, quantified by accuracy and balanced accuracy over all pixels. Here balanced accuracy is included to account for the imbalance of positive to negative pixels. Results on these metrics are shown in the lower section of Table 1. The model trained with all bands achieves a balanced accuracy (overall accuracy) of 0.77 (0.98) compared to 0.73 (0.97) for the model with just bands 11 and 12, indicating that inclusion of other channels also improves performance at the pixel level.





## 4 Application to monitoring super-emitter sites

A potential application for such a model is in providing automated monitoring of locations where previous emissions are known
to have occurred. To quantify the skill of CH4Net in this setting we produce predictions for all available images during the
2021 validation period at each of the 21 training sites. For this application important scene-level metrics to consider are the
accuracy (the total percentage of correct predictions), false alarm rate (percentage of incorrect alerts), and false negative rate
(percentage of missed alerts). Pixel-level metrics assessing the quality of the generated mask are the accuracy and balanced
accuracy. Results for each site are presented in Table 2. In all cases, these are generated using the optimal predictor set with all
bands (ALL).

At a scene level, high accuracy is observed for a majority of sites, with accuracy greater than $80\%$ for 17 out of 21 sites, and
ranging from 0.6 to 0.79 for remaining sites. False positive rates range from 0.01 to 0.4, and false negative rates from 0.0 to
0.42, though are below 0.2 for a majority of sites.

At a pixel level, accuracy values are greater than 0.97 for all sites. As this is primarily a result of the over-representation of
negative pixels, balanced accuracy is a more informative metric. Balanced accuracy ranges from 0.62 to 1.0, with 18 out of the
21 sites above 0.75.

To better understand the successes and limitations of this approach, we present several case studies, two of locations with
excellent prediction quality (sites $T_7$ and $T_{21}$) and two with poor prediction quality (sites $T_1$ and $T_{11}$).

### 4.1 Case studies: sites $T_7$ and $T_{21}$ (high quality predictions)

For example, consider site $T_7$ where the prediction system has a balanced accuracy score of 0.90, with false positive rate of
0.06 and false negative rate of 0.16 for a site where $39\%$ of scenes in the validation set contain an emission. Figure 4 compares
predictions to the observed values for scene-level classification. Overall predictions are in good agreement with observations,
correctly identifying two emissions early in 2021 followed by a period of high emission activity which subsides towards the
end of the year.

Predictions at site $T_{21}$ provide an example of correct prediction of a single emission event over the course of the validation
year with an additional false positive at the end of the time series. For this site the scene level accuracy is 0.99, false positive
rate 0.01, false negative rate 0.0 and pixel level balanced accuracy 0.72.

### 4.2 Case studies: sites $T_1$ and $T_{11}$ (low quality predictions)

We next examine two cases with comparatively poor prediction quality. Results for site $T_1$ are the worst out of all locations
with at least one emission during the validation year, with an accuracy of 0.69, false positive rate of 0.35 and false negative
rate of 0.12. A time series of predictions compared to observations is shown in the upper panel of Figure 5. This demonstrates
that the model produces a high number of false positives, particularly through the second half of the year. Closer examination
of individual predictions images indicates that there are three primary sources of false positives. Artifacts in the image (e.g.,
Fig. 6(a)) and thin clouds (e.g., Fig. 6(b)) produce occasional false positives throughout the time series. During the second



**Table 2.** CH4Net performance evaluated on all available images at the 23 super-emitter sites for 2021. It is emphasized that sites $T_6$ and $T_{17}$ are not included in the training set.

| Site | longitude | latitude | % positive | ACC (scene) | FPR | FNR | ACC (pixel) | balanced ACC (pixel) |
|------|-----------|----------|-----------|-------------|-----|-----|-------------|----------------------|
| $T_1$ | 53.6367 | 39.49687 | 17.0% | 0.69 | 0.35 | 0.12 | 0.98 | 0.81 |
| $T_2$ | 53.77274 | 39.52148 | 0.0% | 0.96 | 0.04 | 0.0 | 1.0 | 1.0 |
| $T_3$ | 53.77903 | 39.52137 | 0.0% | 0.86 | 0.14 | 0.0 | 0.99 | 0.99 |
| $T_4$ | 53.74292 | 39.4739 | 1.0% | 0.86 | 0.14 | 0.0 | 0.99 | 0.94 |
| $T_5$ | 53.78836 | 39.46428 | 1.0% | 0.79 | 0.22 | 0.0 | 0.99 | 0.67 |
| $T_6$ | 53.77502 | 39.4616 | 38.0% | 0.92 | 0.04 | 0.15 | 0.98 | 0.78 |
| $T_7$ | 53.77921 | 39.45965 | 39.0% | 0.9 | 0.06 | 0.16 | 0.99 | 0.79 |
| $T_8$ | 53.68117 | 39.44955 | 0.0% | 0.99 | 0.01 | 0.0 | 1.0 | 1.0 |
| $T_9$ | 53.76506 | 39.36045 | 23.0% | 0.7 | 0.37 | 0.09 | 0.98 | 0.75 |
| $T_{10}$ | 53.83516 | 39.38584 | 0.0% | 0.92 | 0.08 | 0.0 | 1.0 | 1.0 |
| $T_{11}$ | 53.87509 | 39.35498 | 8.0% | 0.95 | 0.01 | 0.5 | 0.99 | 0.62 |
| $T_{12}$ | 54.23498 | 38.85515 | 15.0% | 0.88 | 0.1 | 0.27 | 0.99 | 0.81 |
| $T_{13}$ | 54.20931 | 38.57959 | 0.0% | 0.92 | 0.08 | 0.0 | 1.0 | 1.0 |
| $T_{14}$ | 54.20049 | 38.55747 | 37.0% | 0.81 | 0.24 | 0.11 | 0.98 | 0.83 |
| $T_{15}$ | 54.20393 | 38.51871 | 0.0% | 0.82 | 0.18 | 0.0 | 0.99 | 0.99 |
| $T_{16}$ | 54.19769 | 38.50798 | 0.0% | 0.97 | 0.03 | 0.0 | 1.0 | 1.0 |
| $T_{17}$ | 54.19764 | 38.49393 | 10.0% | 0.96 | 0.05 | 0.0 | 0.99 | 0.97 |
| $T_{18}$ | 54.02832 | 38.33078 | 16.0% | 0.73 | 0.25 | 0.42 | 0.99 | 0.71 |
| $T_{19}$ | 54.03149 | 38.36017 | 0.0% | 0.62 | 0.38 | 0.0 | 0.98 | 0.98 |
| $T_{20}$ | 53.89857 | 37.90825 | 16.0% | 0.84 | 0.16 | 0.17 | 0.99 | 0.76 |
| $T_{21}$ | 53.91623 | 37.9286 | 1.0% | 0.99 | 0.01 | 0.0 | 1.0 | 0.72 |
| $T_{22}$ | 53.92431 | 37.92913 | 23.0% | 0.88 | 0.04 | 0.41 | 0.99 | 0.67 |
| $T_{23}$ | 53.92702 | 37.71665 | 0.0% | 0.6 | 0.4 | 0.0 | 0.98 | 0.98 |

half of 2021 multiple false positives are produced coinciding with a bright surface artifact visible in both the RGB and MBMP images (e.g., Fig. 6(c)). It is possible that this is a methane emission source, however, it is not labelled as such during the manual labelling as either the wind speed is too low to produce a clear plume or alternatively the emissions are weak with only the area immediately at the source detectable with the limited detection capability of Sentinel-2.

Site $T_{11}$ is an example of a site with multiple false negatives. For this location, the scene accuracy is 0.95, with a false positive rate of 0.01 however the false negative rate is the highest for all sites at 0.5. The prediction time series for this site is shown in the lower panel of Figure 5. Here the false negatives appear to arise in cases with heterogeneous background (which also often results in an increase in false positives). This is consistent with recent work indicating that the detection capability of Sentinel-2 is significantly lower in cases with a strongly heterogeneous background (Gorroño et al., 2023).





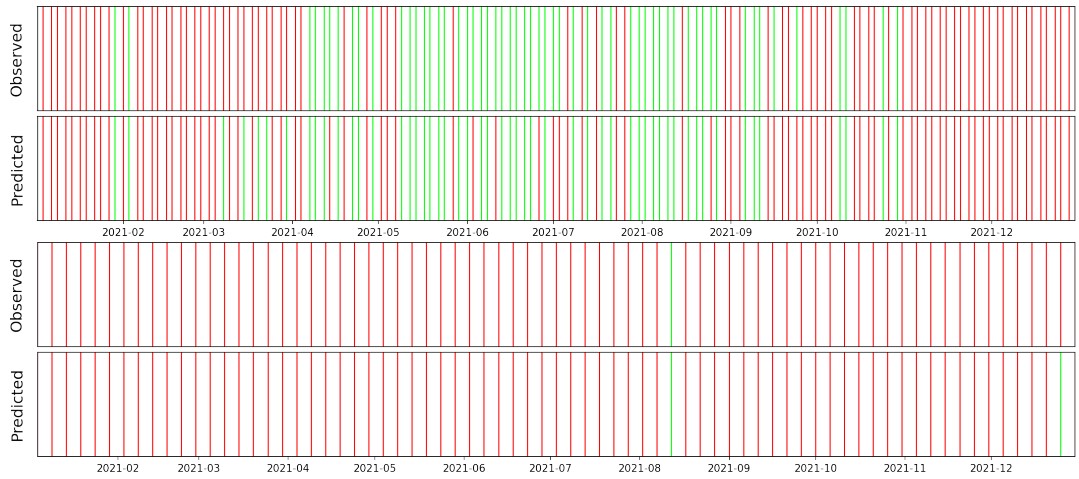

**Figure 4.** Time series of predictions for sites $T_7$ (top) and $T_{21}$ (bottom) over the validation year (2021). Green (red) lines indicate that a plume was (not) observed or predicted. Observed ground truth values are shown in the upper time series and CH4Net predictions on the lower time series.

## 5 Skill at new sites

An interesting question is whether the trained CH4Net model can generate skillful predictions at locations unseen at training time. To evaluate skill in this scenario two locations were held out at random (sites $T_6$ and $T_{17}$), and CH4Net applied to generate predictions at these locations for 2021.

  For site $T_6$, CH4Net achieves a scene-level accuracy of 0.92, false positive rate of 0.04 and false negative rate of 0.15. At a pixel level, the balanced accuracy is 0.78 (Table 2). A time series of predictions is shown in 7 (upper panel). This demonstrates

overall excellent skill, with two sporadic emissions correctly identified at the beginning of the time series, followed by a lengthy period of continuous emissions, which are largely correctly detected before the site ceases emitting towards the end of the year.

  A more detailed view of predictions at a pixel scale is shown in Figure 8. This shows the observation mask compared to prediction overlaid on the RGB imagery for every available Sentinel-2 image in 2021. Both the occurrence and morphology of each plume is largely well captured, though several false positives and false negatives are observed during the period of high

activity in the middle of the year.

  For the second held out site, $T_{17}$ CH4Net achieves an accuracy of 0.96, false positive rate of 0.05, false negative rate of 0.0 and pixel level balanced accuracy of 0.97. Similar to the analysis for site $T_6$ above, the time series of scene level classifications is shown in Figure 7 (lower panel) with pixel level predictions shown in Figure 9.





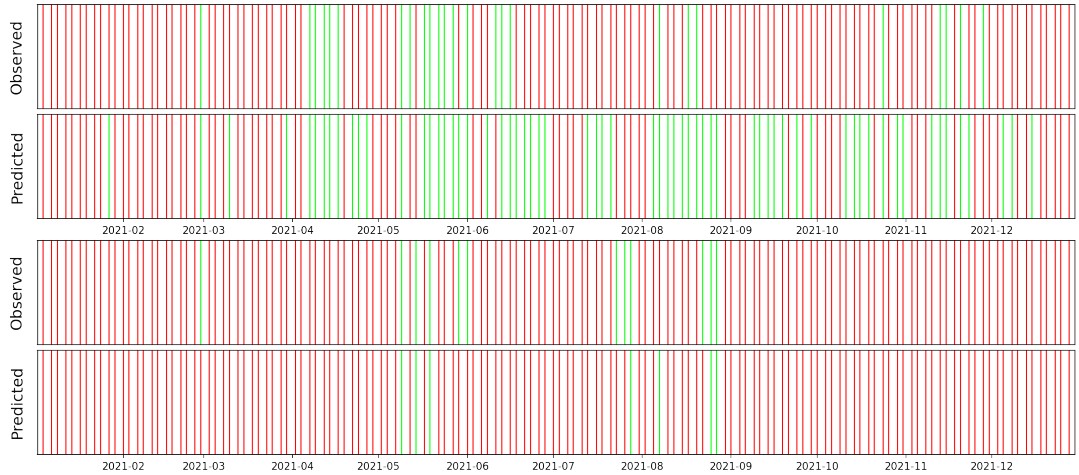

**Figure 5.** Time series of predictions for sites $T_1$ (top) and $T_{11}$ (bottom) over the validation year (2021). Green (red) lines indicate that a plume was (not) observed or predicted. Observed ground truth values are shown in the upper time series and CH4Net predictions on the lower time series.

For this site, CH4Net produces two false positives early in the year, however, subsequently correctly predicts all positive and negative events. Further analysis on a pixel level indicates that the morphology of the predicted masks is of high quality.

These results indicate that CH4Net has potential for application to other new sites in the same semi-arid region for which training data are not available. It is emphasized however that these results only investigate the two held out locations which are both from the same region as the training set, and further more detailed study is required to rigorously quantify prediction skill on both a larger set of new sites and sites with different background characteristics.

## 6 Conclusions

We have implemented CH4Net, the first fully automated system for monitoring known methane super-emitter sites and produced the first large scale dataset of methane plumes in Sentinel-2 imagery. Model skill was assessed on multiple scene-level and pixel-level metrics, demonstrating that overall predictions are of high quality, though several sources of false positives and false negatives remain to be addressed. We further demonstrated that this system can successfully be applied to generate skillful predictions at new locations unseen at training time. These results offer promise for implementing ongoing tracking of known sources to mitigate emissions and provide early warnings when an event is observed.





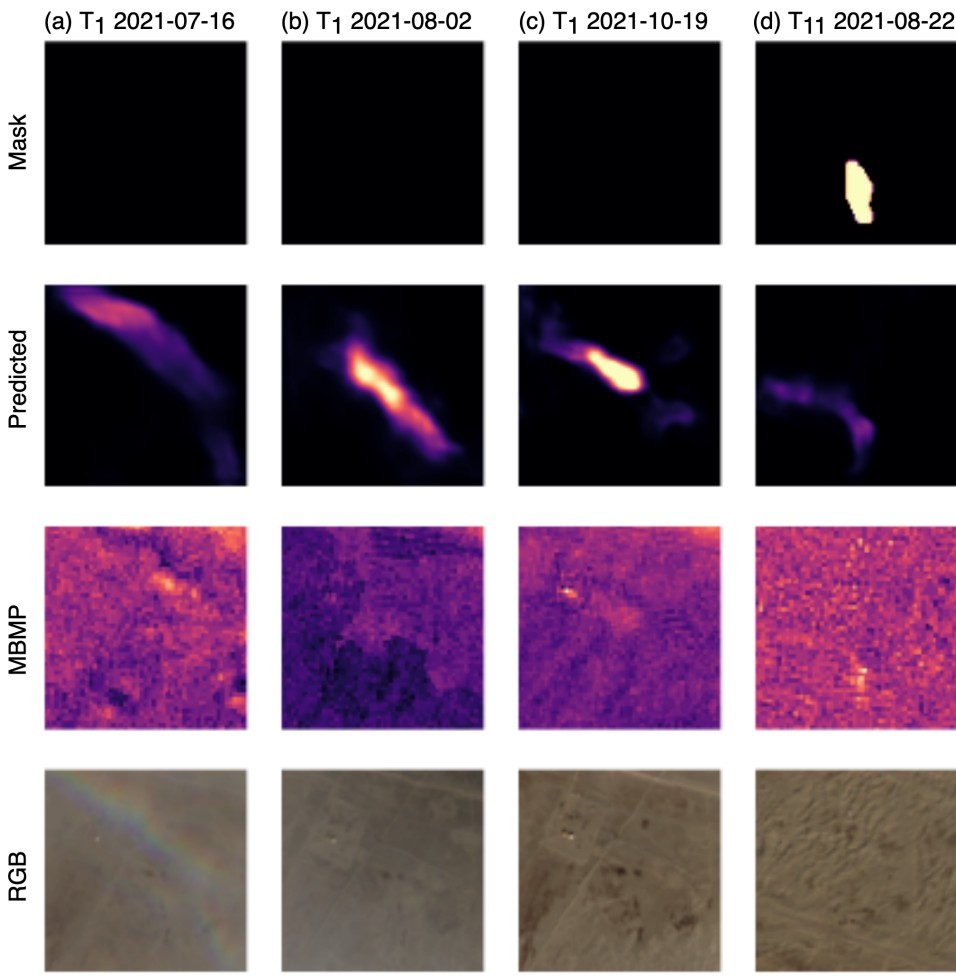

**Figure 6.** Examples of false positives and negatives for sites $T_1$ and $T_{11}$, showing: (a) false positive at site $T_1$ resulting from image artefact, (b) false positive at site $T_1$ resulting from thin cloud (not easily visible in the RGB window), (c) false positive at site $T_1$ resulting from potential low intensity methane source and (d) false negative at site $T_{11}$ resulting from strongly heterogeneous background.

In contrast to existing methods for methane plume detection in Sentinel-2 images (Varon et al., 2021; Ehret et al., 2022; Irakulis-Loitxate et al., 2022), this model requires only a single pass to generate predictions and is fully automated. This creates a significant advantage in allowing large volumes of data to be processed without requiring costly manual verification.



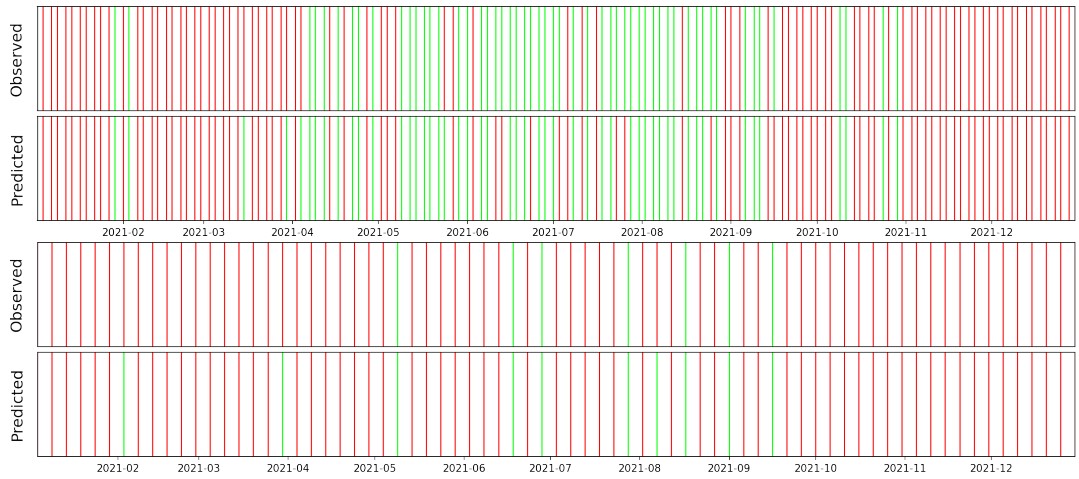

**Figure 7.** Time series of predictions for sites $T_6$ (top) and $T_{17}$ (bottom) over the validation year (2021). Green (red) lines indicate that a plume was (not) observed or predicted. Observed ground truth values are shown in the upper time series and CH4Net predictions on the lower time series.

Further work is required in several areas to extend these promising results. A priority for future work is to collect further data over new areas and test whether CH4Net is suitable for application to other semi-arid locations. A current shortcoming of this work is that the output of CH4Net provides only a binary mask as opposed to quantifying the methane concentration at each pixel. Direct prediction of this quantity would allow for both emission occurrence and volume to be monitored. Additionally, we hope to extend this dataset to further emission locations outside of Turkmenistan, for example to Algeria, Libya, and Iran, which have high emissions and similar surface characteristics.

*Code and data availability.* Code and hand-annotated masks are available at https://github.com/annavaughan/CH4Net. Sentinel-2 data are available from Sentinel Hub https://www.sentinel-hub.com/.

*Author contributions.* A.V designed the study, implemented the code, labelled the dataset, conducted the experiments and wrote the first draft. All authors contributed to the analysis of results and final version of the paper.

*Competing interests.* The authors declare no competing interests.





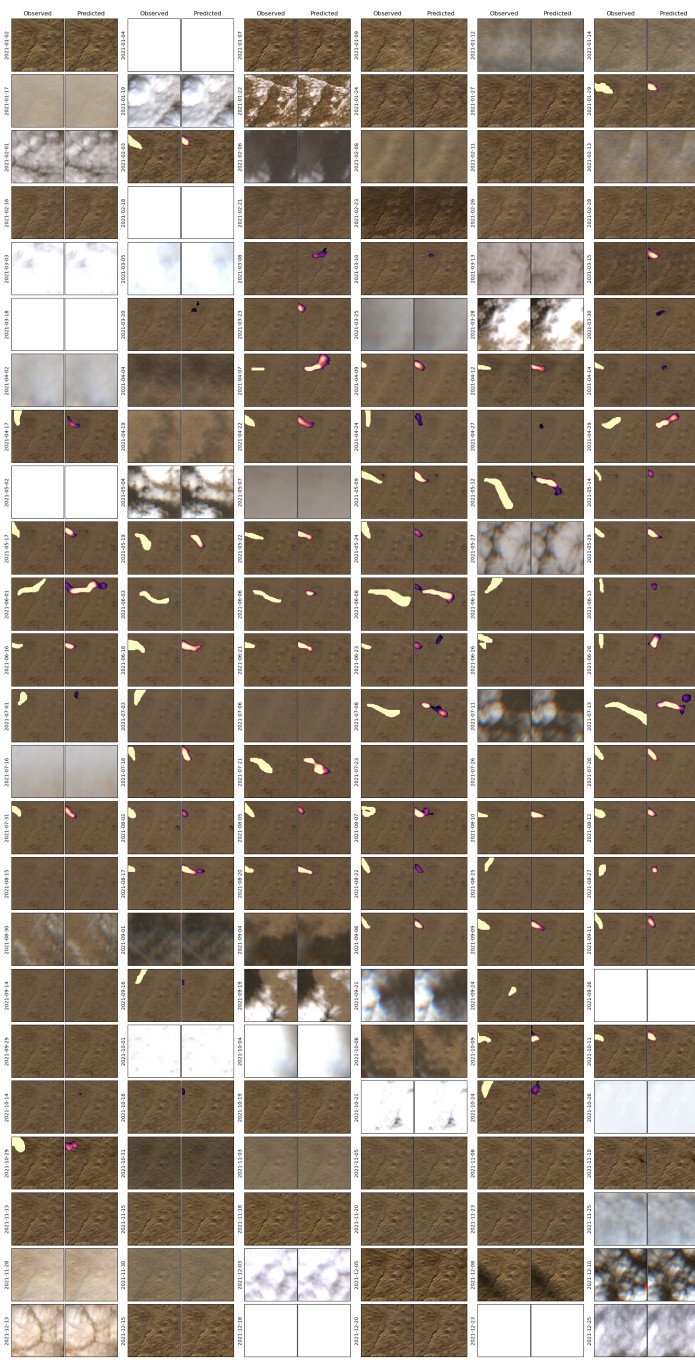

**Figure 8.** CH4Net pixel-level predictions for every image over site $T_6$ during 2021. For each time-step the observed mask (left) and probabilistic prediction (right) are shown overlaid on the RGB image.



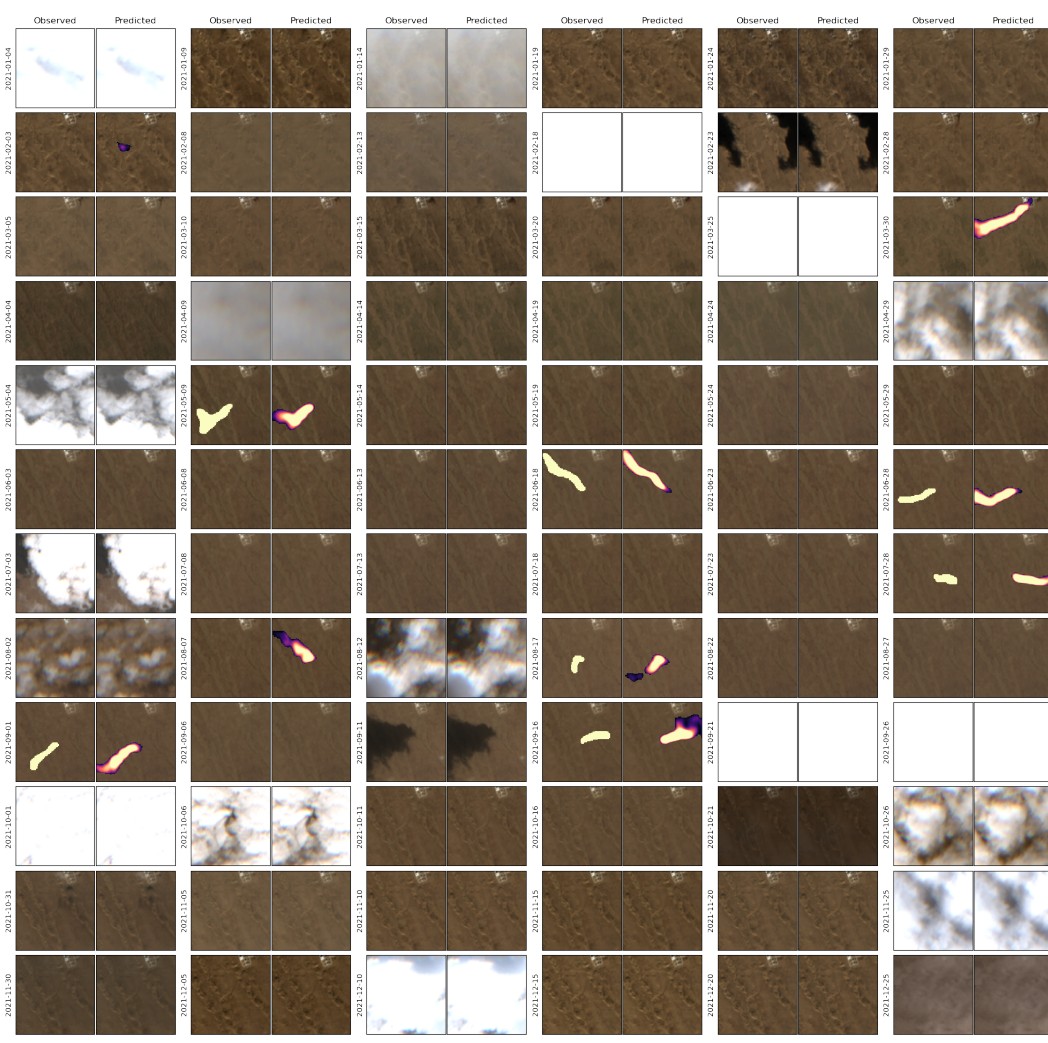

**Figure 9.** CH4Net pixel-level predictions for every image over site $T_1 7$ during 2021. For each time-step the observed mask (left) and probabilistic prediction (right) are shown overlaid on the RGB image.



*Acknowledgements.* Early stages of this project were funded as part of European Space Agency 3CS grant of Trillium Technologies reference `Starcop 1-2022-00380`. Authors gratefully acknowledge the support of the Trillium team and ESA technical officer. A.V. acknowledges the UKRI Centre for Doctoral Training in the Application of Artificial Intelligence to the study of Environmental Risks (AI4ER), led by the University of Cambridge and British Antarctic Survey, and studentship funding from Google DeepMind. G.M.-G. and L.G.-C. have been partially supported by the Spanish Ministry of Science and Innovation (project PID2019-109026RB-I00, ERDF) and the European Social Fund.



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
