# Peer review of "CH4Net: a deep learning model for monitoring methane super-emitters with Sentinel-2 imagery"

_EGUsphere, 2023_

## Author Comment (AC3)

**General response**

We would like to thank all the reviewers for their detailed comments. We have made a number of changes which we believe have significantly improved the manuscript. Below we summarize major changes, then respond individually to each reviewer's comments. We thank the reviewers again for taking the time to provide such insightful feedback.

Major changes:
- Overall paper narrative: We feel that the objective of the paper was not made clear enough in the first manuscript. Our objective with this model is to develop a monitoring system for known super-emitter locations. We originally included a brief experiment exploring how the model generalizes to two new sites (Section 5 of the previous draft), however agree with the reviewers comments that a much more detailed analysis is needed for this application. Given that our overall aim is the monitoring of known sites, we have decided to delete this section from our previous draft and focus solely on the primary monitoring application, mentioning extension to new areas as a topic for future work. This is something we are very interested to explore however we feel is best approached as a separate study as opposed to a small experiment tacked on to the end of this work.

- Baseline: given the suggestions of the reviewers we have included comparison to a multiband-multipass baseline which is currently considered the state of the art method for methane plume detection in Sentinel-2 images.

- Metrics and validation: the reviewers provided great advice on improving the strength of the validation. We have made a number of changes here. We now split the dataset into train, test and validation partitions. We have also rethought how metrics in Table 1 are calculated. Initially we reported all metrics on a balanced subset of all positive images together with a randomly sampled set of negative images which we believed provided more easily interpretable results. After careful thought we also now include all images (positive and negative) in the validation set, as opposed to a subset stratified by plume presence as before. We believe that this provides more useful results to assess for suitability for application in the real world where methane plumes are comparatively rare events. In Table 2 we originally included accuracy, false positive rate and false negative rate as the scene level metrics, we agree with the reviewers that precision and recall are also informative. These were not included in the original manuscript as F-score and recall are undefined for cases where no plumes are observed in the validation period. We therefore retain the original metrics and add precision and recall for cases where at least one plume is observed in the validation period. For pixel-wise metrics we have added intersection over union as suggested.

**Reviewer 1**

General comments:

The paper "CH4Net: a deep learning model for monitoring methane super-emitters with Sentinel-2 imagery" proposes a new method for monitoring and detection of Methane emissions from Sentinel-2 data with a convolutional neural network segmentation model (UNet). The authors collect a large dataset of Sentinel-2 images for locations of known Methane super-emitters (10k+ images) and manually annotate Methane plumes in 925 of those images.

In contrast to existing methods, this paper proposes a fully automatic system for Methane classification/segmentation that operates on single Sentinel-2 images (no manual intervention, time-series, or reference images).

The proposed method is evaluated with good results on two different tasks: Monitoring of methane emissions at known locations and detection of Methane emissions at unseen locations (of known Methane emitters).

The chosen use-case is well motivated as remote detection and monitoring of Methane emissions are powerful tools to mitigate the release of greenhouse gas emissions. The paper is well written and describes the proposed methodology in sufficient detail. However, there are some central issues around the machine learning methodology and the presentation of results.

Specific comments:

**R1Q1** The Methane plume masks are manually annotated using the multi-band multi-pass approach of Varon (2021). Judging from the examples in Figure 2, annotating the plumes depends to a significant degree on knowledge about the exact location of the emitter in the image as well as the annotator. It would be valuable to collect an additional set of annotations for the same locations from at least one other annotator. This would help to quantify the uncertainty on the labels (e.g., by computing the intersection over union of different annotators) and provide an upper bound for possible model performance.

Thank you for this comment. While we agree that collecting masks from multiple annotators would indeed be ideal it would require significant resources to complete which are unfortunately unavailable to us (labeling the full dataset took several hundred hours). We acknowledge that this is a shortcoming of the work and have added a discussion in the future work section at line 188 discussing this.

As a reader I would also appreciate a negative example (no plume) in Figure 2.

Two examples of negative images have been added to Figure 2.

**R1Q2** The presented detection and monitoring use-cases both assume known Methane emitter locations. Therefore, the benefits of a model that operates on single images only is not clear to me. Approaches like MBMP or time-series analyses are very helpful to detect Methane in Sentinel-2 imagery (illustrated by the use of MBMP for labeling in this work). Why not let the model take advantage of this additional information? Given the spectral bandwidths of the Sentinel-2 MSI, it is very difficult to detect Methane in single images. To the best of my understanding, using multiple images would be compatible with the proposed use-cases, as they are restricted to known emitter locations.

We agree with the reviewer that to date multitemporal approaches have been necessary to achieve any detection ability in Sentinel 2 imagery. We choose a single image approach because (a) it is significantly easier to implement, deploy and maintain and (b) it significantly improves over the existing multi-temporal approaches.

Multi-temporal approaches require obtaining a previous image over the same location that is cloud and methane free. In our experience this is quite tedious and cumbersome to implement. Even with cloud masks available, it is frequently necessary to manually select the reference image due to errors in cloud and cloud shadow detection, changes in the surface or presence of snow. We also note that the advantages conferred by adding an extra pass are largely available to the model during training. As the model is trained on multiple scenes not containing a plume the information about background for each scene is learned at training time. For this reason, while an interesting avenue for future work, adding a longer time series of images will not necessarily improve model performance.

As suggested by Reviewer 3 we compare our single pass approach with thresholding of MBMP image finding that it substantially outperforms this baseline. As we publish our dataset and the trained model, we hope that future work can explore the development of a machine learning based multi-temporal method and assess whether this improves model performance. We have added a note outlining this reasoning to the methods section at line 84.

**R1Q3** My central issue with this work is the split of the dataset for training, validation, and testing. Currently, the authors use three splits: The train set, which contains data from 2017-2020 and all but two locations. A "validation" set of the same locations as the train set with data from 2021, and the "held out dataset" with data from the two remaining locations in 2021. The validation dataset is then used to evaluate the Methane monitoring use-case, while the "held out dataset" is used for the Methane detection use-case. It is unclear if the validation dataset is also used to perform for model selection and hyperparameter tuning (as is common in the machine learning literature) or if a sub-set of the training data is used for this purpose. In any case, I strongly discourage the use of the same locations for training, validation and testing as it allows the model to overfit on seen locations.

Instead, I suggest a random train/evaluation/test split by locations for the detection use case, and by locations and time for the monitoring use-case. The models should be tuned based on the train and validation data while only reporting final metrics based on the test set.

Thank you for this suggestion, we agree that this implementation was poorly thought through. As noted in the major changes section we have significantly revamped the analysis and now perform the train/test/validation split as

- Train: all images from 2017-2020 excluding the test set
- Test: a held out randomly subsampled selection of 256 train images stratified by plume presence
- Validation: all images from 2021

After careful thought we also now include all images (positive and negative) in the validation set, as opposed to a subset stratified by plume presence as before. We believe that this provides more useful results to assess for suitability for application in the real world where methane plumes are comparatively rare events. This description is updated in the text at line 88.

R1Q4 I appreciate the many different metrics and figures describing the evaluation results. However, most of them do not properly take the highly imbalanced nature of the data (most pixels do not contain Methane plumes) into account. To provide a more nuanced analysis I would welcome the addition of a balanced accuracy metric for the classification task, and Intersection over Union scores for the plume segmentation task.

Thank you for these suggestions, we have added these metrics to Tables 1 and 2. Additionally, several metrics are already included also take into account the (extremely) unbalanced nature of the task, specifically precision and recall which have now been added to Table 2 and balanced accuracy for the pixel level metrics.

R1Q5 The detection use-case focuses on locations of known emitters, in my view a more appropriate test of detection capabilities would be to "scan" a larger area (perhaps multiple adjacent Sentinel-2 images, containing some known emitter locations) to look for plumes. This test would also highlight the proposed model's large-scale data processing capabilities without need for human intervention or time-series data.

We agree with the reviewer that having a model able to process a full S2 tile and correctly detect the existing plumes would be very useful, however we are tackling a different problem in this paper. By design our model lacks this capacity as it is trained on a limited set of locations and will hence not generalize to cover large heterogeneous areas. We reiterate that this paper focuses on monitoring known locations; i.e. we have shown that a

model trained with past data can generalize to future data over the same locations or to data from a similar location.

In our view, in order to have such a model, it must be  trained on a much larger corpus of heterogeneous images (images from different locations and biomes). We would be very interested in tackling this problem in the future but we believe it is currently beyond the scope of this work (mainly because of the massive work of data labeling that it would require). This is a very interesting avenue for future work and we have added a section to future work discussing this at line 196.

**R1Q6** Furthermore, I am curious about the role of temporal patterns in the data that might be correlated with plume presence. For example, are plumes more frequently observed in summer vs. winter?

We agree with the reviewer that this would be an interesting point, the publication of (Irakullis-Loritxate et al 2022) covers the temporal evolution of emissions in the same study area and period (section results, subsection "temporal evolution of emissions"). In particular they highlight that, in the 2017-2020 period "2018 was the year with the fewest detected emissions, while 2020 has been the year with the most detected emission plumes, double the number detected in 2018 (see Figure 4 and Table 1). This relationship also holds when we normalize the number of emissions by the number of clear-sky observations in each period." Checking if these or other patterns hold when detections are produced by our automatic model would be interesting nevertheless in our opinion these results would be very anecdotal since we have a limited number of sites and our validation set only comprises one year.

Technical corrections:

> The link to the code points to missing webpage.
> This has now been fixed.
> Table 2: the "% positive" column is inconsistent with the performance metrics (there is percent sign there)
> This column denotes the percent of images containing plumes. We have updated the table caption to clarify this.
> It is my understanding that the plume masks are binary, but in some figures more than 2 values seem to be present (perhaps an interpolation issue at the plume/background border (e.g., Figs. 3 and 6))
> The masks are probabilistic, each pixel therefore takes a value from 0-1. We have added a not explaining this at line 104.

> It would be interesting to compare the "constructed classification" model with a dedicated classifier. Given your dataset it could be straightforward to train a binary

classification model that directly predicts the presence of a sizable plume in each image. Would this model perform better at detection that the UNet?

We agree that this is an interesting idea. In this scenario we wished to produce masks as these are necessary for downstream quantification of emission rates. We have added a note that such a hybrid classification/segmentation system is an interesting avenue for research in the future work section at line 193.

**Reviewer 2**

The paper presents CH4Net - a methane plume detection and segmentation neural network model trained on Sentinel-2 imagery in Turkmenistan. Turkmenistan is known for having optimal observing conditions for remote sensing technology that relies on solar backscatter (bright, homogeneous, arid region), so CH4 net results in this paper can be seen as a bounding result for plume detections via Sentinel-2. The authors went to great lengths to create a training set and should be applauded for that effort. I have a few comments on the manuscript in regards to how they summarize their results, which I outline below:

**R2Q1** Line 22. You say that PRISMA and EnMAP provide the most accurate concentration retrievals. What does this mean? In terms of single-sounding precision? That's precision, not accuracy. Also - please be clear what tasking means - they each are limited to X number of X by X km2 tasks per day that are split across a variety of hyper spectral applications.

Hyperspectral retrievals with PRISMA and EnMAP produce more sensitive retrievals (they are able to capture plumes with weaker concentrations of methane). Intuitively, this is because we have more measurements of radiance in the 2100-2350 region, hence, we can better reconstruct the atmospheric spectrum and check if it correlates with the methane absorption signature.

Following the reviewer's suggestion, we have changed this paragraph to be more precise. We also added a reference in the following paragraph to (Sherwin et al 2023) which compares retrievals of different sensors (hyperspectral and multispectral) over controlled methane releases.

**R2Q2**. Line 45 and Point (2) in your introduction. You previously state that the benefit of your approach is that you only need a single overpass as opposed to a time-series, like Ehret. However, if you are splitting your data into train/test that train on one period of time and test on another period of time in the same location, then intrinsically you have added

temporal information into your model. Your model is learning surface features along with plume info, correct?

The difference is that a MBMP model requires a timeseries of images to make a prediction, while CH4Net only requires a single overpass. We have added a note explaining this at line 84.

**R2Q3**. Line 45 and Point (2) in your introduction. What is the motivating use-case for not wanting multiple overpasses to reduce noise? Latency for plume detection? Leak detection? It is not made clear in the manuscript how this is a significant benefit. For example, one could envision a spin-up period where you well characterize surface reflectance features in a region. Once that's initialized, every subsequent overpass of Sentinel-2 would result in a low latency plume detection. So not clear to me the benefit of emphasizing this use-case. Please explain further.

We agree with the reviewer that to date multitemporal approaches have been necessary to achieve any detection ability in Sentinel 2 imagery. We choose a single image approach because (a) it is significantly easier to implement, deploy and maintain and (b) it significantly improves over the existing multi-temporal approaches.

Multi-temporal approaches require obtaining a previous image over the same location that is cloud and methane free. In our experience this is quite tedious and cumbersome to implement. Even with cloud masks available, it is frequently necessary to manually select the reference image due to errors in cloud and cloud shadow detection, changes in the surface or presence of snow. We also note that the advantages conferred by adding an extra pass are largely available to the model during training. As the model is trained on multiple scenes not containing a plume the information about background for each scene is learned at training time. For this reason, while an interesting avenue for future work, adding a longer time series of images will not necessarily improve model performance.

As suggested by Reviewer 3 we compare our single pass approach with thresholding of MBMP image finding that it substantially outperforms this baseline. As we publish our dataset and the trained model, we hope that future work can explore the development of a machine learning based multi-temporal method and assess whether this improves model performance. We have added a note outlining this reasoning to the methods section at line 84.

**R2Q4**. Table 1 and scene-level statistics. Can one back out easily the number of detected plumes vs. the number of total plumes using this summary info? If not, can you please include?

Thank you for this suggestion, we have now included recall in Table 2.

In a similar vein as Reviewer #1 - I am curious as to your model performance if you trained a classification model, e.g., CNN, on this dataset and got similar performance.

We agree that this is an interesting idea. In this scenario we wished to produce masks as these are necessary for downstream quantification of emission rates. We have added a note that such a hybrid classification/segmentation system is an interesting avenue for research in the future work section at line 193.

**R2Q5**. Line 115 and Table 1. Can you please define balanced accuracy in this context? If balanced accuracy = (true positive rate + true negative rate) / 2 for example, then you are still going to get overly optimistic results. For example, assume that 1% of pixels in a scene are plume pixels, then working backwards, a 77% balanced accuracy score would mean that your true positive rate was only 55%: (55+99)/2 = 77%. So why not show these in Table 1 as well? Similar to Reviewer #1's comment - did you try metrics like Intersection over Union? Did they provide similar results?

Thank you for these suggestions, here balanced accuracy is defined as (TPR+TNR)/2. The true positive rate is already included in Table 1 (recall is another name for true positive rate). We have added IoU to Tables 1 and 2 as suggested.

**R2Q6**. Can we see predictions for your high quality examples, like Figure 6? In particular, for T21 - would be interested in seeing the plume mask for the correct prediction vs. the false positive.

We have added a figure (Figure 5) showing pixel-level predictions for a high-quality example. For T21 we no longer observe any false positives (retraining the model using the new train/test/validation set naturally changes the predictions slightly), so we instead chose T17 for this figure as more plumes are observed with two false positives as opposed to T21 with only a single plume in the timeseries.

**Reviewer 3**

The authors describe a CNN-driven plume detection system trained + validated on multispectral Sentinel-1 (repeat) observations of 26 superplume sites in Turkmenistan. The data collection, labeling, data preprocessing and model preparation + training processes are sound, but there are significant issues with the sampling + validation methodology that require additional work and further clarification. Additionally, the authors provide no comparisons to baseline or state-of-the-art approaches. These issues must be addressed in order to provide the reviewer sufficent context to assess the capabilites of their model and the significance of this application.

**R3Q1** My primary concern with this paper is with respect to the impact of spatial bias on the provided results. Specifically, by applying the current training/validation methodology to nearly 1k scenes representing only 26 sites with superplumes, it is highly probable that CH4net system is learning to distinguish labeled regions where plumes have previously occurred within the selected sites from (regions in) non superemitter sites, rather than consistently distinguishing pixels representing CH4 plumes from pixels with no observed CH4 present. The somewhat mixed results on the two hold out sites are inadequate to demonstrate robust plume detection. To demonstrate robust plume detection performance, the authors need to provide additional results where the validation set is spatially disjoint from the training set. A 60/40 train/val split (i.e., all data from 16 sites in the training set vs. all observations from the remaining 10 sites in the val set) should provide roughly similar sampling proportions as their current methodology, and would more effectively capture how well the approach generalizes.

The main results of this paper focus on monitoring known locations with super-emissions; i.e. we show that a model trained with past data can generalize to future data over the same locations. The overall aim was somewhat unclear in the original manuscript, which has been significantly reworked to better communicate the intended use case. Although this model is trained for specific locations, we believe that this is already a significant breakthrough since as it has been shown in other works (e.g. Irakulis-Loritxate 2022) emissions are persistent over certain locations; with this model we envision a system that, when a new location is added, we can label past data, retrain the model and use it to produce notifications of new plumes on incoming Sentinel-2 acquisitions over that location. This would be very useful to verify that leaks have been permanently fixed and to notify the emitters if this is not the case.

We agree that the results in Section 5 of the original manuscript, while interesting, were not sufficient to demonstrate robust generalization. As our primary interest in this work is developing a model to monitor known sites we opt to remove Section 5 from the results and instead add this experiment to the future work section at line 195.

**R3Q2** Another concern is that separating superplumes from background enhancements is often achievable with simple image processing methods (e.g., applying a threshold to a band ratio product). The pixelwise concentrations of typical superplume enhancements often (dramatically) exceed the (numerical magnitudes of pixels representing) nominal background enhancements observed in many remote sensing GHG products. The authors provide no baseline comparisons with alternative baseline approaches (e.g., thresholding the MBMP images or the ratio between bands 11/12, with a threshold determined by plume vs. background pixel magnitudes), so the reviewer cannot assess whether a CNN is truly necessary for this detection problem. At minimum, results on the scenewise plume detection task using a basic "straw man" approach should be provided to demonstrate that the detection problem is nontrival.

Thank you very much for this suggestion, we have added a comparison to the MBMP thresholding technique to the results, finding that our model substantially improves plume detection performance.

**R3Q3** I would suggest one additional minor change wrt Table 1: the authors should replace the aggregate pixel level accuracy / balanced accuracy scores with the pixelwise FPR/FNR (or TPR/TNR) averaged across the validation scenes. Because plumes are relatively rare, the vast majority of pixels are background (negative class) pixels, so if a classifier predicts all pixels in all scenes are not plumes, the average accuracy will approach 100%. While the balanced accuracy is slightly more informative, it does not specify whether prediction errors are false positives or false negatives.

Thank you very much for this suggestion, after considering the advice of all three reviewers we have added pixelwise IoU as the most suitable pixelwise metric to address the class imbalance issue.

---

## Author Response (AR2)

**Author response**

We thank the reviewers for their suggestions, and are pleased that they feel the majority of the issues with the manuscript have been resolved. Responses to the final two minor points are detailed below in blue. Thank you again to all the reviewers for their insightful contributions throughout the peer review process.

Minor point raised by Reviewer #1: "To the best of my understanding, the authors use the "test" split for hyperparameter tuning/model selection and the "validation" split to report results. I would suggest to switch the naming of these splits, in line with common practice in the machine learning field."

Thank you for noticing this, we have made this change throughout the manuscript.

Reviewer #2 still has an issue with your statement that unlike other methods CH4Net does not require any time series of images: "The authors response doesn't quite answer the issue. Both your model and the multi-pass retrieval models have temporal information. The difference is that Ehret/Varon/MBMP/etc models put temporal information into the retrieval, while you are putting temporal information into the learning algorithm. For example, CH4Net relies on multiple images (in time) of a few select sites in order to create a robust model. So yes, at the time of prediction, all you need is a single overpass. However, in order to make a successful prediction, you need temporal information. I would hope the authors address this in the manuscript."

Thank you for this comment. We agree that this is an important distinction to make and have made the following alterations to the manuscript.
- Line 53: we have noted the distinction between our method and previous methods by adding the explanation "In contrast to previous methods, CH4Net learns background characteristics of the sites by processing multiple passes over each location during training without the need for a time series of previous images, reference image, or manual verification step."
- Updated references to CH4Net only requiring a "single image" with "single image at test time" throughout the manuscript.